# Comparison of the Diagnostic Value of MRI and Whole Body ^18^F-FDG PET/CT in Diagnosis of Spondylodiscitis

**DOI:** 10.3390/jcm9051581

**Published:** 2020-05-22

**Authors:** Corinna Altini, Valentina Lavelli, Artor Niccoli-Asabella, Angela Sardaro, Alessia Branca, Giulia Santo, Cristina Ferrari, Giuseppe Rubini

**Affiliations:** 1Nuclear Medicine Unit, Interdisciplinary Department of Medicine, University of Bari Aldo Moro, Piazza Giulio Cesare, 11–70124 Bari, Italy; corinna.altini@hotmail.it (C.A.); valentina.lavelli@gmail.com (V.L.); alessia9130@gmail.com (A.B.); giuliasanto92@gmail.com (G.S.); giuseppe.rubini@uniba.it (G.R.); 2Nuclear Medicine Unit, AOU Policlinic “A. Perrino”, 72100 Brindisi, Italy; artor.niccoliasabella@asl.brindisi.it; 3Section of Radiology and Radiation Oncology, Interdisciplinary Department of Medicine, University of Bari Aldo Moro, Piazza Giulio Cesare, 11–70124 Bari, Italy; angela.sardaro@uniba.it

**Keywords:** spondylodiscitis, spine infection, MRI, ^18^F-FDG PET/CT

## Abstract

Spondylodiscitis is a spine infection for which a diagnosis by a magnetic resonance imaging (MRI) is considered the most appropriate imaging technique. The aim of this study was to compare the role of an ^18^F-fluorodeoxyglucose positron emission tomography/computed tomography (^18^F-FDG PET/CT) and an MRI in this field. For 56 patients with suspected spondylodiscitis for whom MRI and ^18^F-FDG PET/CT were performed, we retrospectively analyzed the results. Cohen’s κ was applied to evaluate the agreement between the two techniques in all patients and in subgroups with a different number of spinal districts analyzed by the MRI. Sensitivity, specificity, and accuracy were also evaluated. The agreements of the ^18^F-FDG PET/CT and MRI in the evaluation of the entire population, whole-spine MRI, and two-districts MRI were moderate (*κ* = 0.456, *κ* = 0.432, and *κ* = 0.429, respectively). In patients for whom one-district MRI was performed, ^18^F-FDG PET/CT and MRI were both positive and completely concordant (*κ* = 1). We also separately evaluated patients with suspected spondylodiscitis caused by *Mycobacterium tuberculosis* for whom the MRI and ^18^F-FDG PET/CT were always concordant excepting in 2 of the 18 (11%) patients. Sensitivity, specificity, and accuracy of the MRI and ^18^F-FDG PET/CT were 100%, 60%, 97%, and 92%, 100%, and 94%, respectively. Our results confirmed the ^18^F-FDG PET/CT diagnostic value in the diagnosis of spondylodiscitis is comparable to that of MRI for the entire spine evaluation. This could be considered a complementary technique or a valid alternative to MRI.

## 1. Introduction

Spondylodiscitis is an infection of the vertebral body or disc that can extend to contiguous soft tissues [1]. Its incidence is rising due to increased life expectancy. It mainly affects men aged between 50 and 70 years [1,2]. It is often associated to the presence of debilitating conditions, such as endocarditis, diabetes mellitus, septic arthritis, urinary tract infections and indwelling catheter infections, malignancy, and spinal surgery [1,2]. The most frequent district involved is the lumbar spine, followed by the dorsal tract and the cervical tract [2]. The principal causes of spondylodiscitis are pyogenic agents, most commonly *Staphylococcus aureus*, followed by *Escherichia coli*. Less commonly, spondylodiscitis is caused by non-pyogenic agents, such as *Mycobacterium tuberculosis*, *Brucella*, *fungi*, and *parasites* [1,3]. 

The diagnosis of spondylodiscitis is based on a clinical suspect for the presence of symptoms, such as focal back pain, fever and/or neurological deficit associated with nonspecific laboratory findings [2,4,5]. Imaging findings are also fundamental for diagnosis and magnetic resonance imaging (MRI) is considered the most accurate technique for the early detection of spondylodiscitis [4,6,7].

More recently, ^18^F-fluorodeoxyglucose positron emission tomography/computed tomography (^18^F-FDG PET/CT) was proven to be a useful multimodality imaging method to study infectious and other benign disease, including spondylodiscitis [8,9].

The aim of this study was to compare the role of an ^18^F-FDG PET/CT and MRI in the diagnosis of spondylodiscitis.

## 2. Experimental Section

We retrospectively analyzed patients who underwent a whole-body ^18^F-FDG PET/CT for the suspicion of spondylodiscitis, performed from April 2013 to October 2018. There were 105 patients (74 men, 31 women; mean age 63 years; range: 18–90 years) for whom ^18^F-FDG PET/CT was performed. Only 56/105 patients for whom MRI was performed before ^18^F-FDG PET/CT were included in the analysis, whereas the other 49/105 were excluded. All the MRIs were performed in an average of 8 days before the ^18^F-FDG PET/CT (range: 3–15 days) and included at least one spinal district. 

All patients had already given their consent for the use of their data for clinical research. Our Institutional Review Board does not require the Ethical Committee’s approval for the review of the patients’ files. 

The MRI examinations of the spine were acquired on a 1.5 T scanner (ACHIEVA, Philips Healthcare, Amsterdam, Netherlands). The choice of which and how many spinal districts to scan was driven by clinical suspicion. The MRI scan protocol consisted of a T1-weighted (T1-W), Turbo Spine Echo (TSE), T2-Weighted (T2-W), and Short Time Inversion Recovery-Weighted (STIR-W). The T1-W sagittal, axial, and coronal scans were performed after the contrast agent was administered to all patients. The total duration of the examination was 20–25 minutes without considering post-processing. The MRI findings indicative of spondylodiscitis were decrease signal intensity from the disc and adjacent vertebral bodies on T1-W images, increase signal intensity from the disc and adjacent vertebral bodies on T1-W, increase signal intensity on T2-W images (due to edema), and loss of endplate definition on T1-W. The gadolinium enhancement of the discs, vertebrae, and surrounding soft tissue helped to differentiate the infective lesions from degenerative changes (Modic type 1 abnormalities) or neoplasm [10].

An integrated PET/CT scanner Discovery IQ (GE, Healthcare Technologies, Milwaukee, WI, USA) was used for obtaining results. Patients were recommended to fast for six hours before the acquisition of the PET/CT. Before the ^18^F-FDG injection, the blood sugar levels were measured and the optimal value was <140 mg/dL. We proceeded with the intravenous injection of 2.5–3 MBq/kg of ^18^F-FDG using a venous line and the patients were allowed to stay in the room for approximately 60 minutes. Whole-body PET/CT acquisitions were performed by placing patients in a supine position with their arms raised above their heads. PET images were obtained in a three-dimensional mode and were analyzed qualitatively and semi-quantitatively with a Multivol PET/CT program of Advantage™ workstation (GE, Healthcare Technologies, Milwaukee, WI, USA). PET findings were analyzed by evaluating the different ^18^F-FDG uptakes between the soft tissue infection and the bone marrow.

In all patients, the final diagnosis of spondylodiscitis was confirmed or excluded on the basis of resolution or significant improvement of constitutional symptoms (back pain and/or fever), laboratory, such as C-reactive protein (CRP), erythrocyte sedimentation rate (ESR), and white blood cells (WBCs), and instrumental (e.g., MRI and biopsy) follow-ups were performed for at least 6 months.

Sensitivity, specificity, and accuracy of the MRI and ^18^F-FDG PET/CT were calculated. Cohen’s κ was also applied to evaluate the agreement between the MRI and ^18^F-FDG PET/CT in the entire population analyzed and in subgroups who performed whole-spine, two-districts and one-district MRI. Analyses were performed by MedCalc® statistical software version 2020 (MedCalc Software Ltd., Ostend, Belgium).

## 3. Results

The demographic and clinical characteristics of the 56 patients are shown in Table 1.

None of patients were started on any antibiotic therapies before the imaging techniques performance. 

The sensitivity, specificity, and accuracy for ^18^F-FDG PET/CT were 92%, 100%, and 94%, and for MRI were 100%, 60%, and 97%, respectively. 

In agreement with clinical and instrumental findings, final diagnosis of spondylodiscitis in 51/56 (91%) patients was confirmed, whereas in 5/56 (9%) patients, it was ruled out.

^18^F-FDG PET/CT showed pathological uptake in the spine and correctly identified 44/56 (79%) patients with spondylodiscitis. Equally, ^18^F-FDG PET/CT showed no significant uptake in the spine and therefore, correctly ruling out spondylodiscitis in 12/56 patients (21%). The MRI identified 53/56 patients (95%) with spinal infection and ruled out infection in 3/56 patients (5%). ^18^F-FDG PET/CT had a false-negative in 4/56 patients (7%). MRI had a false-positive in 2/56 patients (4%) with a final diagnosis of severe degenerative disc disease.

Table 2 reports the *κ* values for the agreement results.

Table 3 reports the MRI and ^18^F-FDG PET/CT results for all patients and subgroups.

Figure 1 reports an example case of spine infection detected by both MRI and ^18^F-FDG PET/CT. Figure 2 reports a case of discordance between the MRI and ^18^F-FDG PET/CT. 

In the subgroup of patients for whom whole-spine MRI was performed, 14/19 patients (74%), had the same lesions where the two techniques were identified. In 4/19 patients (21%), the MRI identified lesions in two districts, whereas the ^18^F-FDG PET/CT only in one. In 1/19 patients (5%), the ^18^F-FDG PET/CT identified one lesion more than MRI.

In the subgroup of patients for whom two-districts MRI was performed, seven of nine patients (78%) had the same lesions which the two techniques identified. In one of nine patients (11%), MRI identified lesions in two districts whereas the ^18^F-FDG PET/CT in only one. In one of nine patients (11%), the ^18^F-FDG PET/CT identified one lesion more than the MRI.

In the subgroup of 19 patients for whom the one-district MRI was performed, the ^18^F-FDG PET/CT and the MRI were both positive and completely concordant for lesions identified that were: cervical in 1/19 (5%), dorsal in 5/19 (26%), and lumbar in 13/19 (69%).

The average value of Standardized Uptake Value (SUVmax) was 6.7 (SD ± 3.2).

The results concerning patients with *Mycobacterium tuberculosis* (TBC) etiology are reported separately in Table 4. 

## 4. Discussion

Diagnosis of spondylodiscitis is a combination of clinical, laboratory, and radiological findings. Symptoms and clinical signs of spondylodiscitis are non-specific and include severe back pain, fever and, in one-third of cases, neurological deficit such as leg weakness, paralysis, sensory deficit, radiculopathy, and loss of sphincter control [2,11]. For this reason, spondylodiscitis is difficult to diagnose because laboratory results are also nonspecific and not always present; they could include elevated levels of CRP, increased ESR, and elevated WBC [1]. 

Diagnosis of certainty associated with the detection of the etiological agent can be obtained by invasive investigations, such as biopsy, but this is associated with significant risks and side effects. For this reason, it is necessary to be able to have non-invasive diagnostic methods with repeatable and high performance value [2,12].

Generally, the first imaging modality required for suspected spinal infection, despite its low sensitivity and specificity, are plain radiographs [2,13,14,15,16]. Usually, the signs are not present on radiographs until two to eight weeks after the onset of symptoms and may not be evident in patients with severe degenerative disc disease [2,15,16]. Nowadays, MRI is the imaging procedure most used due to its high sensitivity in the diagnosis of infection and in assessing the extent of disease [4,17]. The main MRI findings for spondylodiscitis are impairment of the intervertebral discs associated with disc space narrowing or possible epidural involvement and increased contrast enhancement in the spine [4,6]. Contrast-enhanced MRI is the method of choice in clinical practice due to its advantages including high contrast resolution, high sensitivity for soft tissue, absence of ionizing radiation exposure, and the possibility to show evidence of bone marrow abnormalities [2,5]. MRI cannot be performed in patients with cardiac implantable electronic devices, cardioverter defibrillators, or cardiac resynchronization devices; even patients with metallic implants cannot undergo the process. The high costs and long run times for whole-spine examination do not make it a suitable examination for all patients. MRI diagnostic performance decreases in follow-up evaluation and in post-operative infection of the spine [2,17,18,19,20,21,22,23]. These drawbacks, over time, lead many to look for other methods that could replace an MRI, mostly when it is contraindicated, not available, or doubtful [2,24,25]. 

Over the years, ^18^F-FDG PET/CT, a whole-body technique, has shown advantages in the clinical and therapeutic management of oncological disease and, simultaneously, its usefulness in the management of several non-oncological disease has been proven [26]. However, different from MRI, ^18^F-FDG PET/CT exposes patients to ionizing radiation even if updated protocols minimize radio exposure. In patients with spondylodiscitis, ^18^F-FDG PET/CT allows for differentiation of degenerative and infectious abnormalities found on an MRI [4,27]. It is considered positive when ^18^F-FDG uptake is higher than bone marrow uptake in adjacent vertebrae or soft tissue around the spine. Therefore, unlike morphological imaging, ^18^F-FDG PET/CT highlights glycidic metabolism, which may already be increased in the early stages of infection [4,5].

The diagnostic value of an MRI and ^18^F-FDG PET/CT in our study is in line with current literature and confirm the higher sensitivity of an MRI and its relatively lower specificity than ^18^F-FDG PET/CT (100% vs. 92% and 60% vs. 100%, respectively). The two techniques showed similar accuracy (94% and 97%). The high sensitivity of an MRI allows the identification of almost all spondylodiscitis lesions, but the lower specificity is indicative of the possibility of misunderstanding lesions due to inflammatory or degenerative spondyloarthropathy, recent vertebral fractures, postoperative inflammation, or bone tumors. The high specificity of ^18^F-FDG PET/CT improves the interpretation of ambiguous MRI images [9].

Previous studies that investigated the value of an MRI in diagnosing spondylodiscitis showed results similar to our study, such as sensitivity of 82–96%, specificity of 85–93%, and accuracy of 81–94% [5,28,29,30]. 

Even though this imaging modality is considered the gold standard for spinal infection diagnoses, in 2015, a prospective study of 26 patients who underwent an MRI and ^18^F-FDG PET/CT reported encouraging results for the use of an ^18^F-FDG PET/CT as a valid alternative in the evaluation of patients with suspected spondylodiscitis that could not perform an MRI. In particular, Fauster et al. found sensitivity and specificity of 83% and 88% for ^18^F-FDG PET/CT and 94% and 38% for MRI, respectively [4]. Similar results were reported in a recent meta-analysis that ^18^F-FDG PET/CT has a superior accuracy than MRI (97% vs. 81%) in detecting spondylodiscitis, as well as high sensitivity (95% vs. 85%) and specificity (88% vs. 66%) [31]. Smids et al. analyzed a group of 75 patients with clinical suspicion of spondylodiscitis and found that ^18^F-FDG PET/CT had higher sensitivity and accuracy in the early diagnosis of spondylodiscitis than MRI, especially when performed in the first two weeks of the onset of symptoms [5]. 

Our study included a homogeneous group of 56 patients with suspected spondylodiscitis, even though the comparison for the entire spine was not performed for all patients. For this reason, we also evaluated the agreement of the two techniques for all of the lesions and it always resulted moderately except for patients with a single-spine district MRI, for whom the agreement was complete and absolute. The differentiation in groups, based on the district studied from the MRI, was necessary to compare overlapping data and therefore to identify differences in numbers and locations of the infection, which can involve multiple spine sites.

Discordant results were found in 6 of 56 patients for whom the MRI was positive, but the ^18^F-FDG PET/CT did not reveal any lesions. In four of them, the diagnosis of spondylodiscitis was confirmed and false-negative ^18^F-FDG PET/CT results were probably due to a weak infection characterized only by epidural involvement, whereas the low ^18^F-FDG uptake may have been related to the different immune reaction caused by the various etiological agents [22,32,33]. In the remaining two patients, spondylodiscitis was finally not confirmed by follow-up and the false-positive MRI results were due to the presence of severe degenerative disc disease [22].

Differences among patients in whom the MRI and ^18^F-FDG PET/CT were concordant in the identification of the suspected lesions were observed. In patients for whom the whole-body MRI was performed, MRI identified one lesion more than the ^18^F-FDG PET/CT in four patients, whereas the ^18^F-FDG PET/CT detected one lesion more than the MRI in only one patient. Similarly, in the subgroup for whom two-districts MRI was performed, it identified one lesion more than the ^18^F-FDG PET/CT in one patient, whereas the ^18^F-FDG PET/CT detected one lesion more than the MRI in only one patient. The higher number of lesions identified by the MRI was related to the presence of edema-like changes in the end plates and to epidural abscesses. The patients in whom additional ^18^F-FDG PET/CT lesions were found, psoas abscesses were detected. For all the additional lesions found by the two techniques, the final diagnosis of spondylodiscitis was confirmed.

*Mycobacterium tuberculosis* is one of the etiological agents of spondylodiscitis, but it can differ from bacterial disease for clinical and imaging. In these patients, the back pain and neurological deficit onset occurred only later in chronic forms [34]. Imaging-guide vertebral biopsy is a gold standard for differentiating tubercular and pyogenic spondylodiscitis. MRI and ^18^F-FDG PET/CT were helpful for evaluating the extension of the infection [35]. The main MRI findings, suggestive for tubercular spondylodiscitis, indicated sparing of the intervertebral disc in the early stages of infection, loss of vertebral body cortical definition, multiple vertebral involvement, and the presence of muscular (paraspinal, psoas) abscesses [34,35]. Tuberculous spondylodiscitis ^18^F-FDG PET/CT commonly presents as ^18^F-FDG uptake lesions. However, in some cases, a cold abscess is characterized by moderate ^18^F-FDG uptake in the cortex and low uptake in the center [36]. 

In our study, we described separate patients with tuberculous spondylodiscitis. In most of them, the MRI and ^18^F-FDG PET/CT results were both of positive and in concordance in identifying lesions due to their typical imaging presentation at two techniques. The remaining two patients’ results were discordant. The false-positive MRI result was the same as described in the entire population and therefore was due to degenerative end-plate abnormalities. Similarly, the false-negative ^18^F-FDG PET/CT result was previously described and may be related to low-virulence bacteria.

Despite the encouraging results, some limitations of our study should be reported. First of all, we analyzed a small sample size in line with the literature. Other limitations were the heterogeneous groups of pathogens and the difference in the numbers of spine districts examined by the MRI. 

Our results show the complementary role of ^18^F-FDG PET/CT and MRI in a spondylodiscitis diagnosis. The combination of ^18^F-FDG PET/CT and MRI detected lesions in 100% of patients with spondylodiscitis, identifying the site and the extent of the disease and correctly guiding the therapeutic choice [2,4,37,38]. 

## 5. Conclusions

Our results confirmed the high diagnostic value of ^18^F-FDG PET/CT in the diagnosis of spondylodiscitis in comparison with MRI. Given the increasing evidence of the diagnostic value of ^18^F-FDG PET/CT, it could be proposed as a possible alternative to MRI, especially when MRI is contraindicated, non-diagnostic, or inconclusive. The agreement between these two techniques suggests their complementary role in selected and more complicated patients. 

## Figures and Tables

**Figure 1 jcm-09-01581-f001:**
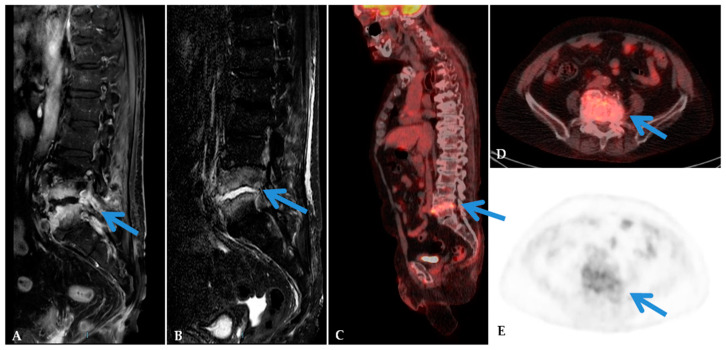
MRI and ^18^F-FDG PET/CT in a 55-year-old man with back pain, fever, and with positive microbiology culture (*Staphilococcus aureus*). Sagittal MRI images showed in T1-Weighted (**A**) and Short Time Inversion Recovery (**B**) sequences pathological signal in the L4-L5 intervertebral disc and bone marrow edema (arrows). ^18^F-FDG PET/CT images (**C**–**E**) showed pathological uptake in L4–L5 (Standardized Uptake Value 6.7) (arrows). The final diagnosis of spondylodiscitis was confirmed.

**Figure 2 jcm-09-01581-f002:**
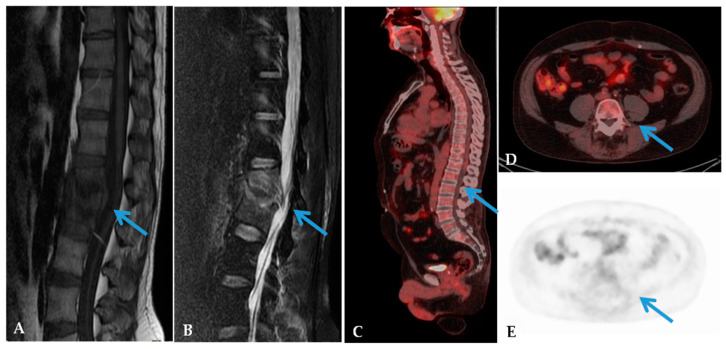
MRI and ^18^F-FDG PET/CT in a 62-year-old man with back pain and low-grade fever. Sagittal MRI T1-Weighted (**A**) and T2-Weighted (**B**) images showed low T1 and high T2 signal at the L1–L2 end plates (arrows). ^18^F-FDG PET/CT images (**C**–**E**) showed no significant FDG uptake in the spine (arrows). These findings confirmed severe degenerative disc disease.

**Table 1 jcm-09-01581-t001:** Baseline characteristics.

Variable	Number (Percentage)
**Total Number of Patients**	**56**
Sex	
Male	39 (70%)
Female	17 (30%)
Median age (years)	63 (18–90)
Symptoms	
Fever	35 (62%)
Back pain	38 (69%)
None	18 (32%)
Etiological agent	52 (93%)
*Mycobacterium tuberculosis*	18 (35%)
*Staphilococcus aureus*	14 (27%)
*Escherichia coli*	4 (8%)
*Haemophilus influenzae*	4 (8%)
*Brucella*	3 (6%)
*Propionibacterium acnes*	2 (3%)
*Streptococcus epidermidis*	2 (3%)
*Pseudomonas aeruginosa*	1 (2%)
*Candida albicans*	1 (2%)
*Aspergillus*	1 (2%)
*Staphilococcus capitis*	1 (2%)
*Granulicatella elegans*	1 (2%)
Not found	4 (7%)
MRI district study	
One	19 (34%)
Cervical	1 (5%)
Dorsal	5 (26%)
Lumbar	13 (69%)
Two	12 (21%)
Cervical/dorsal	1 (8%)
Dorsal/lumbar	5 (42%)
Cervical/lumbar	1 (8%)
Lumbar/sacral	5 (42%)
Whole-spine	25 (45%)

**Table 2 jcm-09-01581-t002:** Agreement results in all patients and MRI subgroups.

		*κ*	95% Confidence Interval
All patients	56	0.456	0.11–0.80
Subgroup I (whole-spine MRI)	25/56 (45%)	0.432	0.01–0.85
Subgroup II (two-district MRI)	12/56 (21%)	0.429	0–1.00
Subgroup III (one-district MRI)	19/56 (34%)	1	0.79–1.00

**Table 3 jcm-09-01581-t003:** Results in all patients and in MRI subgroups.

	MRI (+)	MRI (+)	MRI (−)	MRI (−)
	**^18^F-FDG PET/CT (+)**	**^18^F-FDG PET/CT (−)**	**^18^F-FDG PET/CT (+)**	**^18^F-FDG PET/CT (−)**
All patients (*n* = 56)	47	6	0	3
Subgroup I (whole-spine MRI)	19	4	0	2
Subgroup II (two-districts MRI)	9	2	0	1
Subgroup III (one-district MRI)	19	0	0	0

Note: MRI (+), ^18^F-FDG PET/CT (+), positive concordance; MRI (+/−), ^18^F-FDG PET/CT (− /+), discordance; MRI (−), ^18^F-FDG PET/CT (−), negative concordance.

**Table 4 jcm-09-01581-t004:** Results in all patients affected by *Mycobacterium tuberculosis* and MRI subgroups.

	MRI (+)	MRI (+)	MRI (−)	MRI (−)
	^18^F-FDG PET/CT (+)	^18^F-FDG PET/CT (−)	^18^F-FDG PET/CT (+)	^18^F-FDG PET/CT (−)
All patients (*n* = 18)	16	2	0	0
Subgroup I (whole-spine MRI)	5	2	0	0
Subgroup II (two-districts MRI)	3	0	0	0
Subgroup III (one-district MRI)	8	0	0	0

Note: MRI (+); ^18^F-FDG PET/CT (+): positive concordance. MRI (+/−); ^18^F-FDG PET/CT (− /+): discordance. MRI (−); ^18^F-FDG PET/CT (−): negative concordance.

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
