# Peer review of "Comparison of the Diagnostic Value of MRI and Whole Body 18F-FDG PET/CT in Diagnosis of Spondylodiscitis"

_jcm, 2020, doi:10.3390/jcm9051581_

Round 1

Reviewer 1 Report

Review comments for JMC796464

The authors comparatively analyzed sensitivity, specificity and accuracy of MRI and 18F-FDG PET/CT for the diagnosis of 66 spondylodiscitis.

The sensitivity, specificity and accuracy of MRI and 18F-FDG PET/CT were 100%, 75%, 97% and 91%, 100% and 92%, respectively. The authors concluded that the 18F-FDG PET/CT is comparable to MRI for the entire spine evaluation and can be considered a complementary technique or a valid alternative to MRI.

This is an interesting regarding 18F-FDG PET/CT for diagnosis of spondylodiscitis. Before resubmitting the article, I request some major and minor changes as follows.

Major point:

  • The most critical point of this article is that timings taking MRI and 18F-FDG PET/CT were quite different in some cases (The authors described 3-36 days in M & M section). Infection sometimes progresses quite rapidly even within 1 week, especially without antibiotics (The authors also described “None of patients started an antibiotics therapy before imaging techniques performance” in M & M section). It is necessary to exclude the cases with long-period of the time between MRI and 18F-FDG PET/CT.
  • Another critical point of this article is major infection of this study was M. Tuberculosis (22/66, 37%). The MRI findings and the clinical courses are completely difference to those of S. aureus or E. coli. Furthermore, cold abscess of M. Tuberculosis shows different MRI and 18F-FDG PET/CT findings. It is difficult to analyze these different pathogeneses together. M. Tuberculosis should be differently analyzed with other bacteria.

Ann Nucl Med. 2005 Sep;19(6):515-8.

Cold Tuberculous Abscess Identified by FDG PET

Skeletal Radiol. 2017 Jun;46(6):777-783.

Higher fluorine-18 Fluorodeoxyglucose Positron Emission Tomography (FDG-PET) Uptake in Tuberculous Compared to Bacterial Spondylodiscitis

  • The authors have to show the exact standardized uptake values (SUV) for diagnosis of spondylodiscitis by18F-FDG PET/CT.

Minor points;

  • The authors cite two references (8, 9) for the previous non-cancerous studies by 18F-FDG PET/CT in Introduction section. However we can already available the article regarding spinal infection examined by18F-FDG PET/CT as follows. Please cite appropriate references and use for discussion.

Spine (Phila Pa 1976). 2015 Jan 15;40(2):109-13.

Role of 18F-fluoro-D-deoxyglucose PET/CT in Diagnosing Surgical Site Infection After Spine Surgery With Instrumentation

Eur J Nucl Med Mol Imaging. 2020 May;47(5):1287-1301. Diagnostic Performance of 18 F-FDG PET/CT in Patients With Spinal Infection: A Systematic Review and a Bivariate Meta-Analysis

Medicine (Baltimore). 2019 Mar;98(11)

Clinical application of dual-phase F-18 sodium-fluoride bone PET/CT for diagnosing surgical site infection following orthopedic surgery.

Ans so on.

  • Please correct “RM district study” to “MRI district study” in Table 1

Author Response

The authors comparatively analyzed sensitivity, specificity and accuracy of MRI and 18F-FDG PET/CT for the diagnosis of 66 spondylodiscitis.

The sensitivity, specificity and accuracy of MRI and 18F-FDG PET/CT were 100%, 75%, 97% and 91%, 100% and 92%, respectively. The authors concluded that the 18F-FDG PET/CT is comparable to MRI for the entire spine evaluation and can be considered a complementary technique or a valid alternative to MRI.

This is an interesting regarding 18F-FDG PET/CT for diagnosis of spondylodiscitis. Before resubmitting the article, I request some major and minor changes as follows.

Major point:

  • The most critical point of this article is that timings taking MRI and 18F-FDG PET/CT were quite different in some cases (The authors described 3-36 days in M & M section). Infection sometimes progresses quite rapidly even within 1 week, especially without antibiotics (The authors also described “None of patients started an antibiotics therapy before imaging techniques performance” in M & M section). It is necessary to exclude the cases with long-period of the time between MRI and 18F-FDG PET/CT.

Following the reviewer’s suggestion and according to literature (Smids, “A comparison of the diagnostic value of MRI and 18F-FDG PET/CT in suspected spondylodiscitis”), we reviewed our sample and excluded patients who performed MRI and 18F-FDG PET/CT with a interval longer than 2 weeks. Therefore, in our new manuscript 56 patients were evaluated; consequently, all statistical analysis was repeated and reported in the manuscript. New results were similar to the previous ones, so discussion and conclusion were not different.

  • Another critical point of this article is major infection of this study was M. Tuberculosis (22/66, 37%). The MRI findings and the clinical courses are completely difference to those of S. aureus or E. coli. Furthermore, cold abscess of M. Tuberculosis shows different MRI and 18F-FDG PET/CT findings. It is difficult to analyze these different pathogeneses together. M. Tuberculosis should be differently analyzed with other bacteria.

Ann Nucl Med. 2005 Sep;19(6):515-8. Cold Tuberculous Abscess Identified by FDG PET

Skeletal Radiol. 2017 Jun;46(6):777-783. Higher fluorine-18 Fluorodeoxyglucose Positron Emission Tomography (FDG-PET) Uptake in Tuberculous Compared to Bacterial Spondylodiscitis

The reviewer correctly underlines the clinical and imaging differences between tuberculous and pyogenic spondylodiscitis. Following his suggestion, we also separately described results about patients affected by M. Tuberculosis. In the manuscript, the results of this subgroup of patients were reported on Table 4 and a discussion of them was added on page 5, lines 218-233.

  • The authors have to show the exact standardized uptake values (SUV) for diagnosis of spondylodiscitis by18F-FDG PET/CT.

Following the reviewer’s suggestion, the SUVmax were collected and reported in the manuscript (page 3, line 128). Furthermore, a sentence about it was added in Experimental section at page 2, lines 88-90.

Minor points:

  • The authors cite two references (8, 9) for the previous non-cancerous studies by 18F-FDG PET/CT in Introduction section. However, we can already available the article regarding spinal infection examined by18F-FDG PET/CT as follows. Please cite appropriate references and use for discussion.

Spine (Phila Pa 1976). 2015 Jan 15;40(2):109-13. Role of 18F-fluoro-D-deoxyglucose PET/CT in Diagnosing Surgical Site Infection After Spine Surgery with Instrumentation

Eur J Nucl Med Mol Imaging. 2020 May;47(5):1287-1301. Diagnostic Performance of 18 F-FDG PET/CT in Patients with Spinal Infection: A Systematic Review and a Bivariate Meta-Analysis

Medicine (Baltimore). 2019 Mar;98(11) Clinical application of dual-phase F-18 sodium-fluoride bone PET/CT for diagnosing surgical site infection following orthopedic surgery.

Ans so on.

The references 8 and 9 previously mentioned have been replaced with the first two indicated by the reviewer. Furthermore, the reference entitled “Diagnostic Performance of 18 F-FDG PET/CT in Patients with Spinal Infection: A Systematic Review and a Bivariate Meta-Analysis” has been reported in the discussion to explain the clinical significance of results regarding sensitivity, specificity and accuracy (page 4, lines 173-177).

  • Please correct “RM district study” to “MRI district study” in Table 1

Thanks for noticing. We have corrected.

Reviewer 2 Report

This is a useful clinical study which compares CT PET/ MRI for spondylodiscitis and as such is important to clinicians. 

The sample size is small (but discitis is a relatively rare phenomenal) and only 28 patients had a whole spine MRI which could directly compare to the whole body CT PET. 

It is clear, well written with a sensible approach.  A table comparing benefits/ drawbacks of CT PET/ MRI in discitis could be very useful. Authors do need to mention that CT PET has a lot of radiation so not free of side effects - do correct this.

The comparison with other studies needs to be better - at the moment, there is little useful comparison with the other studies mentioned - they are just noted almost as a list. The evaluation of the results needs to be more nuanced - in this study MRI shows a higher sensitivity compared to the CT PET but the specificity is higher with CT PET.  What does this mean in practice and how does this compare with other studies? Do authors think that the use of MRI for particular areas rather than whole spine affected results 

The discussion needs to end with a critique of this study e.g. small numbers/ select population and how this compare with other studies. I think the conclusion may need to be more nuanced and have some clarity as to what this means for clinicians - based on this work, it may be a two step protocol is proposed (as they mention for complex patients.)

The 22 patients who had a TB diagnosis - did they all have a full spine?

Overall a very worthwhile and potentially useful addition to literature on this topic.

Author Response

This is a useful clinical study which compares CT PET/ MRI for spondylodiscitis and as such is important to clinicians.

The sample size is small (but discitis is a relatively rare phenomenal) and only 28 patients had a whole spine MRI which could directly compare to the whole-body CT PET.

It is clear, well written with a sensible approach. A table comparing benefits/drawbacks of CT PET/ MRI in discitis could be very useful. Authors do need to mention that CT PET has a lot of radiation so not free of side effects - do correct this.

As suggested by the reviewer, we added (on page 4, lines 163-164) the following clarification: “However, different from MRI, 18F-FDG PET/CT expose patients to ionizing radiation even if update protocols minimize radioexposure.”

The comparison with other studies needs to be better - at the moment, there is little useful comparison with the other studies mentioned - they are just noted almost as a list. The evaluation of the results needs to be more nuanced - in this study MRI shows a higher sensitivity compared to the CT PET but the specificity is higher with CT PET.  What does this mean in practice and how does this compare with other studies? Do authors think that the use of MRI for particular areas rather than whole spine affected results

Following reviewer’s suggestion, discussion have been improved by insertion of a new paragraph and correcting some sentence at page 4, lines 170-177. On page 5, lines 198-200, we also explain why different MRI subgroups were studied: “The differentiation in groups, based on the district studied from MRI, was necessary to compare overlapping data and therefore to identify differences in numbers and locations of the infection which can involve multiple spine sites.”

The discussion needs to end with a critique of this study e.g. small numbers/ select population and how this compare with other studies. I think the conclusion may need to be more nuanced and have some clarity as to what this means for clinicians - based on this work, it may be a two step protocol is proposed (as they mention for complex patients.)

As the reviewer’s suggestion, we added a paragraph about limitations of our study (page 5, lines 234-237): “Despite the encouraging results, some limitations of our study should be reported. First of all we analyze a small sample size even if in line with the literature. Other limitations are the heterogeneous group of pathogens and the difference in the numbers of spine districts examined by MRI”.

Furthermore, we added also a paragraph that can better explain the clinical value of our results (on page 5, lines 173-177): "The high sensitivity of MRI allows to identify almost all spondylodiscitis lesions but the lower specificity is indicative of the possibility of misunderstanding lesions due to inflammatory or degenerative spondyloarthropathy, recent vertebral fractures, postoperative inflammation or bone tumors. On the other hand, the high specificity of the 18F-FDG PET/CT would improve the interpretation of doubtful MRI images”.

The 22 patients who had a TB diagnosis - did they all have a full spine?

Following the changes based on the suggestion of another reviewer, we limited our sample to patients who performed MRI and 18F-FDG PET/CT within 2 weeks each other and analyzed separately patients with tubercular pathogenesis. The additional statistical results of this subgroup of patients was reported in Table 4 and was discussed on page 5, lines 218-233.

Overall a very worthwhile and potentially useful addition to literature on this topic.

Thanks you for the suggestion; the bibliography has been modified and implementing with the following ones:

-Spine (Phila Pa 1976). 2015 Jan 15;40(2):109-13. Role of 18F-fluoro-D-deoxyglucose PET/CT in Diagnosing Surgical Site Infection After Spine Surgery With Instrumentation

-Eur J Nucl Med Mol Imaging. 2020 May;47(5):1287-1301. Diagnostic Performance of 18 F-FDG PET/CT in Patients With Spinal Infection: A Systematic Review and a Bivariate Meta-Analysis

-Ann Nucl Med. 2005 Sep;19(6):515-8. Cold Tuberculous Abscess Identified by FDG PET

-Skeletal Radiol. 2017 Jun;46(6):777-783. Higher fluorine-18 Fluorodeoxyglucose Positron Emission Tomography (FDG-PET) Uptake in Tuberculous Compared to Bacterial Spondylodiscitis

-BMC Musculoskelet. Disord. 2017, 18. Magnetic resonance imaging of bacterial and tuberculous spondylodiscitis with associated complications and non-infectious spinal pathology mimicking infections: a pictorial review.

Reviewer 3 Report

The authors report an interesting study comparing the accuracy of 18F-FDG PET-CT vs. MRI for the diagnosis of spondylodiscitis in a retrospective series of 66 patients. The topic is not original but it could be of interest to the scientific community. Material and Methods sound good, and the results are reported in detail. Nevertheless, I have some minor concerns that, if addressed, could increase the overall value of the paper:

1) The manuscript is not always reader-friendly. For example, the abstract should be entirely rephrased to become easier to read. Also, Results shoudl be reported in a simple way, trying to refer to table or figure and avoiding to provide excessively confusing numbers.

2) On page 2, lines 89-90, the authors reported the following sentence: "In all patients, the final diagnosis of spondylodiscitis was confirmed or excluded on the basis of 89 clinical and instrumental follow up performed for at least 6 months." Please, could you explain which clinical or instrumental criteria were used to confirm or deny the diagnosis?

3) Please, remove the tables and figures from the text.

4) One of the main results of the article is that the accuracy in the diagnosis of spondylodiscitis is 97% for MRI vs. 92% for 18F-FDG PET/CT. The authors should comment on that.   

Author Response

The authors report an interesting study comparing the accuracy of 18F-FDG PET-CT vs. MRI for the diagnosis of spondylodiscitis in a retrospective series of 66 patients. The topic is not original but it could be of interest to the scientific community. Material and Methods sound good, and the results are reported in detail. Nevertheless, I have some minor concerns that, if addressed, could increase the overall value of the paper:

1) The manuscript is not always reader-friendly. For example, the abstract should be entirely rephrased to become easier to read. Also, Results shoudl be reported in a simple way, trying to refer to table or figure and avoiding to provide excessively confusing numbers.

As the reviewer suggested the abstract has been rewritten, even considering the word number limitation.

Also results has been modified and reported in tables as reviewer requested, considering the further results about tuberculosis that have been added. As a consequence of the new rewriting of results, sentences about figures have been edited as follow (page 3, lines 115-116): “Figure 1 reports an example case of spine infection detected by both MRI and 18F-FDG PET/CT. Instead, Figure 2 reports a case of discordance between MRI and 18F-FDG PET/CT.”

2) On page 2, lines 89-90, the authors reported the following sentence: "In all patients, the final diagnosis of spondylodiscitis was confirmed or excluded on the basis of clinical and instrumental follow up performed for at least 6 months." Please, could you explain which clinical or instrumental criteria were used to confirm or deny the diagnosis?

Thanks for the suggestion. On pages 2-3, lines 92-95, we explain the criteria to confirm or exclude diagnosis of spondylodiscitis: “In all patients the final diagnosis of spondylodiscitis was confirmed or excluded on the basis of resolution or significant improvement of constitutional symptoms (back pain and/or fever), laboratory results such as C-reactive protein (CRP), erythrocyte sedimentation rate (ESR) and white blood cell (WBC), and instrumental (e.g. MRI and biopsy) follow-up performed for at least 6 months.”

3) Please, remove the tables and figures from the text.

We are so sorry. In the new version of manuscript, we proceeded to send it separately.

4) One of the main results of the article is that the accuracy in the diagnosis of spondylodiscitis is 97% for MRI vs. 92% for 18F-FDG PET/CT. The authors should comment on that.

Thanks for the suggestion. In the discussion (on page 5, lines 170-177), we added a paragraph in which we discussion the results about diagnostic value of MRI and 18F-FDG PET/CT: “The diagnostic value of MRI and 18F-FDG PET/CT resulted from our study are in line with current literature and confirm the higher sensitivity of MRI and its relatively lower specificity than 18F-FDG PET/CT (100% vs 92% and 60% vs 100% respectively); otherwise the two techniques showed similar accuracy (94% and 97%). The high sensitivity of MRI allows to identify almost all spondylodiscitis lesions but the lower specificity is indicative of the possibility of misunderstanding lesions due to inflammatory or degenerative spondyloarthropathy, recent vertebral fractures, postoperative inflammation or bone tumors. On the other hand, the high specificity of the 18F-FDG PET/CT would improve the interpretation of doubtful MRI images”.

Round 2

Reviewer 1 Report

The authors responded to reviewer’s comments one by one.

I can accept this revised version for the publication of JCM.